# Seroprevalence Assessment of Anti-Varicella Antibodies among Adults in the Province of Florence (Italy)

**DOI:** 10.3390/vaccines12091056

**Published:** 2024-09-16

**Authors:** Angela Bechini, Marco Del Riccio, Cristina Salvati, Benedetta Bonito, Beatrice Zanella, Massimiliano Alberto Biamonte, Mario Bruschi, Johanna Alexandra Iamarino, Letizia Fattorini, Lorenzo Baggiani, Monica Della Fonte, Giovanna Mereu, Paolo Bonanni, Working Group, Sara Boccalini

**Affiliations:** 1Department of Health Sciences, University of Florence, 50134 Florence, Italy; marco.delriccio@unifi.it (M.D.R.); cristina.salvati@unifi.it (C.S.); beatrice.zanella1@libero.it (B.Z.); paolo.bonanni@unifi.it (P.B.); sara.boccalini@unifi.it (S.B.); 2Medical School of Specialization in Hygiene and Preventive Medicine, University of Florence, 50134 Florence, Italy; massimilianoalberto.biamonte@unifi.it (M.A.B.); mario.bruschi@unifi.it (M.B.); johannaalexandra.iamarino@unifi.it (J.A.I.); letizia.fattorini@unifi.it (L.F.); 3Azienda USL Toscana Centro, 50122 Florence, Italy; lorenzo.baggiani@uslcentro.toscana.it (L.B.); monica.dellafonte@uslcentro.toscana.it (M.D.F.); giovanna.mereu@uslcentro.toscana.it (G.M.)

**Keywords:** test ELISA, vaccination coverage, chickenpox, epidemiology, predictor, VZV, adult, immunization

## Abstract

**Background:** Varicella infections follow a benign course in around 90% of cases, with more severe forms occurring in adults. To identify potential pockets of susceptibility and to improve targeted immunization strategies, this study aims to critically assess immunological status by evaluating varicella seroprevalence among adults (18–99 years) in the province of Florence (Italy), nearly a decade after Tuscany introduced the vaccination program. **Methods:** A convenience sample of 430 subjects aged 18 to 94 years (mean age 51.8 ± 18.8 years), stratified by age and sex (53.7% of subjects were female; N = 231), was collected between 2018 and 2019. Sero-analytical analyses were conducted utilizing EUROIMMUN Anti-VZV ELISA (IgG) kits. **Results:** Most of them were of Italian nationality (87.4%; N = 376). Among the 430 tested samples, 385 (89.5%) were positive and 39 (9.1%) were negative. The remaining six sera (1.4%), confirmed as equivocal, were excluded from further analysis. No significant differences were found based on sex (*p*-value = 0.706) or nationality (*p*-value = 0.112). The application of trend tests (Mantel–Haenszel; Kendall Tau-b) showed a significant trend (*p* < 0.024 and *p* < 0.032, respectively), with an increasing probability of finding a positive anti-varicella serological status passing from a lower age group (84.2%) to a higher one (93.0%). By considering the female population aged 18–49 years, the seroprevalence of anti-varicella antibodies was found to be 88.4%, with a susceptibility of 11.6%, highlighting the risk of acquiring infection during pregnancy. **Conclusions:** The introduction of varicella vaccination has had a significant impact on public health in Tuscany and in Italy more generally. However, further efforts should be made to reduce the number of individuals still susceptible in adulthood, with particular attention given to women of childbearing age and the promotion of vaccination through mass and social media and institutional websites.

## 1. Introduction

Varicella (or chickenpox) is one of the most common exanthematous diseases, being highly contagious, and is due to a primary infection by the varicella zoster virus [1,2]. It is estimated that varicella follows a benign course in around 90% of cases [3], with more severe forms occurring in adults, immunocompromised individuals, pregnant women, and infants under one year of age [4,5]. In immunocompetent adults, one of the most serious complications that may occur is pneumonia, while in immunocompromised individualsvery intense, prolonged, and sometimes hemorrhagic rashes appear. In these cases, the infection often takes a malignant course with high fever and mucosal bleeding [6,7]. The World Health Organization estimates that there are approximately 140 million cases of chickenpox annually worldwide [8]. Epidemiological data on chickenpox in Europe are based on mandatory reports from eighteen countries of the Union. The latest report from the European Centre for Disease Prevention and Control (ECDC) dates to 2010, and recorded 592,681 cases of chickenpox in that year. The highest incidence rates were reported in Poland, the Czech Republic, Estonia, and Slovenia [9]. In Italy, the incidence of the disease has practically halved in recent years, going from 180 cases per 100,000 inhabitants in 2003 to 99 cases per 100,000 inhabitants in 2013 [6]. This is mainly due to the availability of a highly effective, safe, and well-tolerated live attenuated vaccine that has been available since 1995 [10,11]. The effectiveness of varicella vaccination has been widely demonstrated in countries adopting a national program [12,13,14,15,16,17,18]. Moreover, in Tuscany, the Universal Varicella Vaccination (UVV) program was introduced in 2008. It involves the administration of two doses of MMRV vaccine, the first at 13–15 months and the second at 5–6 years [19,20].

Hence, this study aims to critically assess immunological status by assessing varicella seroprevalence among adults (18–99 years) in the province of Florence, nearly a decade after Tuscany introduced the vaccination program. Specifically, we aimed to identify potential pockets of susceptibility to enhance targeted immunization strategies.

## 2. Materials and Methods

The current seroprevalence assessment is part of the project “*Progetto Sierologia*” [21,22,23,24,25,26], coordinated by the Department of Health Sciences of the University of Florence in collaboration with Meyer Children’s Hospital and the Local Health Unit Toscana Centro. The local Ethics Committee Tuscany Region—Area Vasta Centro (CEAVC) approved the protocol (project identification code: DSS- UNIFI, n. registro pareri 98/2017). The study was performed in accordance with the Declaration of Helsinki. 

### 2.1. Study Population and Setting for Enrollment

The study population consisted of a convenience sample of the adult general population aged >18 years old living in the province of Florence (Tuscany, Central Italy), stratified by age and sex. To ensure the sample more accurately reflected the demographic composition of the general adult population, it was stratified by sex and age. This stratification allowed the proportions of participants in each subgroup to correspond closely with those in the overall adult population. The Morgagni Health Center located in Florence was the setting for the adult population enrollment during the period April 2018–December 2019. All subjects gave their written consent to join the study, and a blood sample was collected from each participant individually. We excluded subjects not living in the province of Florence, subjects who were immunocompromised or taking immunosuppressive treatments, those who had had an acute infectious disease (including measles, rubella, varicella, hepatitis A, and hepatitis B) in the previous two weeks, and subjects who had received a blood transfusion within the six months prior to the enrolment. All blood samples were identified with a progressive numerical identifier to guarantee anonymization, and then analyzed: they were centrifuged (1600 rpm at 4 °C) for 10 min, and the recovered sera were stored at −20 °C until they could be tested for VZV antibodies. 

### 2.2. Serological Analysis 

Seroanalytical analyses were conducted utilizing EUROIMMUN Anti-VZV ELISA (IgG) kits (EUROIMMUN Medizinische Labordiagnostika AG, Lubeck, Germany), boasting a specificity of 95% and a sensitivity of 100%. The absorbance results of the samples post-spectrophotometer reading (Tecan, Infinite F50—Tecan Trading AG, Tecan Group Ltd, Männedorf, Switzerland) were interpreted both quantitatively—determining the concentration of varicella antibodies—and qualitatively, referencing the cutoff values recommended by EUROIMMUN^®^ (EUROIMMUN Medizinische Labordiagnostika AG, Lubeck, Germany). For the determination of sample concentrations, a calibration curve was constructed for each serological session using Magellan Infinite F50 software (version 7.2). The curve is constructed by measuring the absorbance of four calibrators at different known concentrations (plotted point to point). The concentrations of anti-varicella antibodies were calculated using Magellan™ Software (https://lifesciences.tecan.com/software-magellan (accessed on 11 September 2024)), with the analysis mode set to linear. After obtaining the concentration values for each sample, a qualitative interpretation was performed, referring to the following cut-off values:Positive: anti-varicella antibodies ≥11 IU (International Unit)/mL;Negative: anti-varicella antibodies <8 IU/mL;Borderline: anti-varicella antibodies ≥8 IU/mL and <11 IU/mL.

### 2.3. Statistical Analysis

The serological results were collected in an Excel database and evaluated for anti-varicella antibody titers. A descriptive analysis was conducted to assess the overall varicella seroprevalence according to the socio-demographic characteristics of the study population (sex, age, and nationality). The study population was divided into the following age groups: 18–29 years, 30–39 years, 40–49 years, 50–64 years, and >65 years. Subjects with dual (Italian–foreign) nationality were classified as foreigners. In terms of serological status, the sample was divided into three groups based on the antibody titers: positive, negative, or equivocal. For the analyses aimed at assessing significant differences between serological status and other sociodemographic characteristics, serological samples with equivocal results were removed. To assess any significant differences among the considered sociodemographic groups, the Chi-square test and Fisher’s exact test for the comparison between dichotomous variables were applied. The Mantel–Haenszel and Kendall-tau B trend tests were used to assess the presence of an increasing linear trend between immunological status and age group. The analyses were performed with RStudio (Version: 2023.06.1 + 524. Posit team (2023). Rstudio: Integrated Development Environment for R. Posit Software, PBC, Boston, MA, USA. URL http://www.posit.co/. accessed on 15 June 2024). A significance level of *p* < 0.05 was considered statistically significant.

## 3. Results

The study involved 430 subjects aged 18 to 94 years (mean age 51.8 ± 18.8 years). With regard to gender, 53.7% were female (N = 231). The majority of the study population was composed of Italian subjects (87.4%; N = 376), while 12.6% (N = 54) were foreigners or had dual nationality. Five age groups were considered (18–29 years; 30–39 years; 40–49 years; 50–64 years; >64 years) (Table 1). More than half of the study population were over 50 years old (52.3%; N = 225).

The seroprevalence of all samples was analyzed over six separate serological sessions. Each time, a new ELISA kit was employed and a new calibration curve for each session was determined. No discrepancies were found among the various measured calibration curves. Among the 430 tested samples, 385 (89.5%) were positive and 39 (9.1%) were negative. The remaining six sera (1.4%), confirmed as equivocal, were excluded from further analysis (Table 2).

Subjects with positive results were mainly aged 50–64 (94.6%), male (91.4%), and Italian (91.9%) (Table 2, Figure 1); people >50 years old accounted for 54.0% (208/385) of all positive sera.

No significant differences were found based on sex (*p*-value = 0.706) or nationality (*p*-value = 0.112) (Table 2). Applying further stratification by age confirmed these results (Appendix A). The application of trend tests (Mantel–Haenszel, Kendall Tau-b) showed a significant trend (*p* < 0.024 and *p* < 0.032, respectively), with an increased probability of finding a positive anti-varicella serological status passing from a lower age group (84.2%) to a higher one (93.0%).

The seroprevalence of anti-varicella antibodies was then thoroughly evaluated by sex and age, including subjects up to 49 years old. In all age groups considered, a prevalence of positive antibody titers was observed, both in men and women. From the obtained results, it is observed that up to 39 years of age, a higher percentage of positive subjects were men. In the age group of 40–49 years, however, 90.7% of women had a positive antibody titer compared to 89.1% of men. In women, it is observed that, as age increases, the percentage of positive subjects also increases, ranging from 85.7% among 18–29 year olds to 90.7% among 40–49 year olds. The percentage of negative subjects fluctuates around 9–12% in both the 30–39-year age group and the 40–49-year age group in both sexes. In younger subjects, these percentages increase, with a negative antibody titer found in 17.2% of males and 14.3% of females aged 18–29 years. By considering the female population aged 18–49 years, the seroprevalence of anti-varicella antibodies was found to be 88.4%, with a susceptibility of 11.6% (Table 3).

## 4. Discussion

The aim of this study was to estimate the seroprevalence of anti-varicella antibodies in a large sample of the adult population residing in the province of Florence (Italy). It emerged that 89.5% of the study population presented a positive antibody titer, 1.4% were equivocal, and 9.1% of the analyzed samples tested negative, and therefore are susceptible to the disease. No significant differences were found based on sex (*p*-value = 0.706) or nationality (*p*-value = 0.112). The seroprevalence data concerning sex and age highlight how the percentage of women of childbearing age with a positive antibody titer increases with age. However, in younger age groups, a significant percentage of women remain susceptible: approximately 14% in the 18–29 age group and around 12% in the 30–39 age group. A seroepidemiological survey conducted in 13 Italian regional centers during 2019–2020 recorded a seroprevalence level of 91.9%. The levels were 89.9% in males and 93.3% in females (*p* = 0.0002). This investigation, in accordance with what was found in our study, reports a similar trend regarding the increasing percentage of seroprevalence from the youngest to the oldest population, and no significant differences were observed for sex [27].

It is noteworthy that data from the Tuscany regional surveillance system show how, over the last decade, there has been a significant decrease in both varicella cases and hospitalizations. The number of notifications has indeed dropped from over 12,000 cases per year recorded in the 1990s to less than 1000 cases in the last decade [16]. In the period between 2013 and 2019, the number of cases averaged around 830 per year. The rates of reported varicella cases remained fairly stable between 2014 and 2019, with values ranging between the minimum value of 19.9 cases per 100,000 inhabitants in 2019 and the maximum value of 24.3 cases per 100,000 inhabitants in 2017. In the biennium 2020–2021, the rates show a severe reduction, reaching the historical minimum of 3.2 cases per 100,000 inhabitants in 2021, before increasing again in 2022 (5.1 cases per 100,000 inhabitants). It is reasonable to hypothesize that measures adopted to limit the transmission of SARS-CoV-2, such as social distancing, mask use, and frequent hand washing, also significantly contributed to the reduction in varicella cases, as the virus is also primarily transmitted through the air [28].

Furthermore, in the Tuscany region, vaccination coverage for varicella at 24 months of age (with data available from 2013 onwards for other birth cohorts) has consistently exceeded the national average, following the introduction of the UVV program in 2008.

Until 2017, the gap between the vaccination coverage percentage in Tuscany (80%) and in Italy (30–40%) was significant. Since 2017, due to the recommendation to implement varicella vaccination on a national scale (National Immunization Plan 2017–2019), there has been an increase in vaccination coverage both at the regional and national levels [29,30]. The target recommended by the World Health Organization (WHO) and also shared by the National Immunization Plan 2023–2025 to limit the circulation of varicella in the community and obtain herd immunity is 95% vaccination coverage with two doses [29,31]. As highlighted by the latest data on vaccination coverage in Italy, made available on 31 December 2022 and relating to the 2020 birth cohort, an improvement is observed of vaccination coverage equal to 93.35%, with an increase of 1.27% compared to 2021 [29]. Currently, a national law in Italy has made varicella vaccination mandatory for school attendance (for subjects aged 0 to 16 years), and this requirement has contributed to drive immunization coverage rates in the pediatric population [32].

Furthermore, countries in Europe with UVV programs have typically achieved high vaccination rates. This is evident in several countries. For example, in Spain, before policy change restrictions, regions with UVV programs boasted coverage rates exceeding 95% for the first dose and 86% for the second. However, following restrictions, national coverage dropped from 45% to 2% within two years. Similarly, Greece achieved significant progress with UVV, reaching over 70% coverage for the first dose among 6–7 year olds in 2012. Additionally, Athens reported the successful completion of age-appropriate vaccinations for over 60% of preschoolers. Germany also exemplifies the positive effect of UVV, with vaccination rates steadily increasing since 2006 [33,34]. In Finland, a two-dose varicella vaccination program started in September 2017, and high vaccination coverage for the first dose, ranging from 85% to 87% in 2019–2022, and for the second dose (58%) were registered; after the introduction of varicella vaccination, a marked decline in varicella cases was observed among the eligible children that were the targets of the national program, but also among unvaccinated children <1 year, suggesting the indirect effect of the vaccinations [35]. If we consider both the Italian and European scenarios, Tuscany is found to have one of the highest vaccination compliance rates in Italy [36].

Regarding limitations, our study did not explore potential factors influencing the observed seroprevalence across different groups, such as vaccination status. As a matter of fact, vaccination determinants have not been investigated; therefore, it is not possible to trace the reasons why some subjects are susceptible. For the aim of this investigation, adult sera samples were collected ten years after UVV introduction at the regional level. At this point, only a few cohorts of children have been vaccinated. The high seroprevalence rates in adulthood observed in our sample can be largely explained by the historically high incidence rates of varicella in Italy. Indeed, adults in Tuscany should have acquired their immunity almost exclusively by natural infection due to the exposure to a massive circulation of varicella zoster virus during the previous decades, rather than by vaccination against varicella or herpes zoster. On the other hand, vaccination against herpes zoster was recommended at the national level for adults aged 65 years and for certain high-risk groups by the age of 18 years, but coverage rates are not available for either older people or for high-risk subjects.

The sample size is small, and the stratification was performed after determining the sample size due to convenience, so the results have poor generalizability. Furthermore, our results are not very recent; in fact, they date back to before the COVID-19 pandemic, and therefore subsequent variations due to the impact of the pandemic are not taken into account. However, our findings provide important insights into a population that remains insufficiently protected. In addition, to date, this is the most recent and updated study related to the Florence area. The data on seroprevalence in our study show a high seropositivity rate, but a number of susceptible subjects remain among young individuals. Our results are also confirmed by other seroepidemiological studies investigating the seroprevalence against varicella in university students, healthcare providers, and women of childbearing age [37,38,39,40,41,42,43,44,45,46], underlining the need for future prevention strategies to address this issue and reduce the risk of congenital varicella [47,48,49,50,51,52]. Moreover, underreporting is another problem for the reliability of incidence data, as in Italy the level of under notification for varicella is very high, and we can expect the highest underreporting rates in adults, as reported by Ciofi degli Atti et al. [53]. In addition, a change in the epidemiology of varicella outbreaks has been observed worldwide during the implementation of varicella vaccination programs, and cases occurring in vaccinated subjects have been reported [54,55,56]. Clinical manifestations of varicella among vaccinated people may go unnoticed by healthcare providers less familiar with varicella presentation (mild and atypical in vaccinated persons) [57,58]. Therefore, seroprevalence studies are more reliable than incidence data for a comprehensive understanding of varicella virus circulation and the impact of vaccination strategies, but also for addressing new public health preventive measures.

In brief, the introduction of varicella vaccination has significantly impacted public health in Tuscany and, more broadly, in Italy. However, further advances can be made, especially to achieve optimal coverage rates and reduce the number of individuals still susceptible. Therefore, it is necessary to continue this commitment by improving access to vaccination services, responding to citizens’ requests to clarify any doubts about the effectiveness and safety of vaccines, and improving access to information on vaccination via mass media, social networks, and institutional websites dedicated to vaccination at regional and national levels [59,60,61,62,63]. Finally, by repeating seroprevalence studies over time, it is possible to monitor how the spread of varicella changes in a population in order to formulate new vaccination strategies.

## 5. Conclusions

Standing on our results, we confirm the importance of continuously monitoring varicella antibody seroprevalence rates and strengthening current vaccination programs to guarantee adequate levels of immunity in the adult susceptible population, especially women of childbearing age, to reduce the risk of infection during pregnancy and to promote the interruption of varicella transmission. Many efforts should be made in Italy to increase the availability of vaccine data for the adult population, since vaccine data are not collected in the vaccination electronic register. Awareness on the need to monitor adult immunization coverage is also being raised at the European level through international initiatives involving experts in vaccine-preventable diseases, public health, and immunology [64]. Moreover, the ageing population is pushing for easier access to vaccinations to contrast the low adherence to recommended vaccinations for adults and to increase quality of life, especially for older people [65,66].

## Figures and Tables

**Figure 1 vaccines-12-01056-f001:**
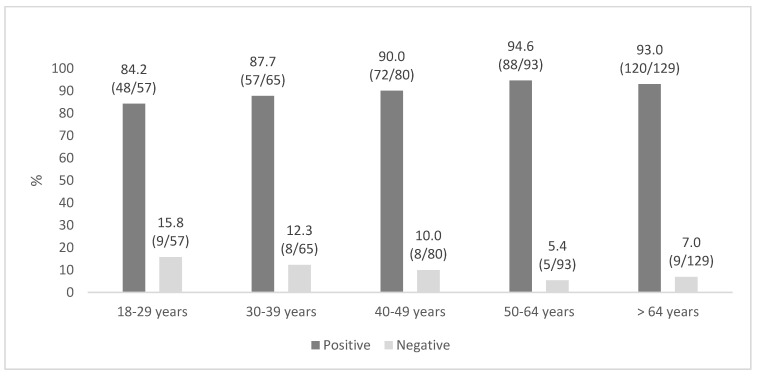
Seroprevalence distribution across age groups. Note: percentages and ratios (n/N) are shown in the figure.

**Table 1 vaccines-12-01056-t001:** Distribution of the study population in terms of age group, sex, and nationality.

Age Group (Years)	Nationality (N)		Sex (N)		Enrolled Subject (N)
	Italian	Non-Italian	Male	Female	
18–29	50	9	29	30	59
30–39	52	14	32	34	66
40–49	64	16	37	43	80
50–64	83	12	46	49	95
>64	127	3	55	75	130
TOTAL	376	54	199	231	430

**Table 2 vaccines-12-01056-t002:** Anti-varicella seroprevalence in the study population divided by age group, sex, and nationality (note: equivocal sera were excluded from the analysis).

Anti-Varicella Seroprevalence				
Variable	Group	Positive % (n/N)	Negative % (n/N)	*p*-Value
Age group (years)	18–29	84.2 (48/57)	15.8 (9/57)	<0.05
	30–39	87.7 (57/65)	12.3 (8/65)	
	40–49	90.0 (72/80)	10.0 (8/80)	
	50–64	94.6 (88/93)	5.4 (5/93)	
	>64	93.0 (120/129)	7.0 (9/129)	
Sex	Female	90.3 (205/227)	9.7 (22/227)	0.706
	Male	91.4 (180/197)	8.6 (17/197)	
Nationality	Italian	91.9 (340/371)	8.1 (31/371)	0.112
	Non-Italian	84.9 (45/53)	15.1 (8/53)	
TOTAL		90.8 (385/424)	8.2 (39/424)	

**Table 3 vaccines-12-01056-t003:** Seroprevalence of anti-varicella antibodies in relation to sex and age up to 49 years.

Age Group (Years)	Sex	Positive N (%)	Negative N (%)	Total N (%)
18–29	M	24 (82.8%)	5 (17.2%)	29 (100%)
	F	24 (85.7%)	4 (14.3%)	28 (100%)
30–39	M	28 (87.5%)	4 (12.5%)	32 (100%)
	F	29 (87.9%)	4 (12.1%)	33 (100%)
40–49	M	33 (89.1%)	4 (10.8%)	37 (100%)
	F	39 (90.7%)	4 (9.3%)	43 (100%)
18–49	M	85 (86.7%)	13 (13.3%)	98 (100%)
	F	92 (88.4%)	12 (11.6%)	104 (100%)

## Data Availability

Data supporting the reported results are available from the corresponding author upon reasonable request.

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
