# Peer review of "Seroprevalence Assessment of Anti-Varicella Antibodies among Adults in the Province of Florence (Italy)"

_vaccines, 2024, doi:10.3390/vaccines12091056_

Round 1

Reviewer 1 Report

Comments and Suggestions for Authors

Nicely constructed paper and this addresses a historical need for varicella immunization as a regional and global goal.

Concerns that I did have while reading the opening sections about population structure of long-term residents of Tuscany vs. migrants (non-long term residents) were answered in the discussion, as was the obvious issue of the data now being close to 6-7 years old.  But, the overall findings and important information on age related differences in seroprevalence are important metrics for public health and national leaders to address, especially since the intervening years post COVID-19 may have altered the status of immunization for susceptible populations.  This would be especially true of children.  If there is a driver that currently helps drive those immunization rates in children, such as requirement for school entry, that could be mentioned in the discussion as an important factor.

One suggested line edit in line 240 replace "Stand" with "Standing"

Author Response

Response to reviewer 1:

Comments and Suggestions for Authors

  1. Reviewer: Nicely constructed paper and this addresses a historical need for varicella immunization as a regional and global goal.

Reply: We would like to thank the reviewer for the positive appraisal of this study.

  1. Reviewer: Concerns that I did have while reading the opening sections about population structure of long-term residents of Tuscany vs. migrants (non-long term residents) were answered in the discussion, as was the obvious issue of the data now being close to 6-7 years old.  But, the overall findings and important information on age related differences in seroprevalence are important metrics for public health and national leaders to address, especially since the intervening years post COVID-19 may have altered the status of immunization for susceptible populations.  This would be especially true of children.  If there is a driver that currently helps drive those immunization rates in children, such as requirement for school entry, that could be mentioned in the discussion as an important factor.

Reply: We thank the reviewer for the considerations provided. We agree that our results, which highlight age-related differences in varicella seroprevalence within the regional population, can be useful for health decision makers, in light of the potential impacts of the COVID-19 pandemic on the immunological status of the population. We agree that this could have affected especially the paediatric population. We thank the reviewer for the suggestion to implement the discussion section. Accordingly, we added the following phrase:

Line 215-217: Currently, in Italy a national law has made varicella vaccination mandatory for school attendance (for subjects aged 0 to 16 years) and this requirement can contribute to drive immunization coverage rates in the pediatric population [19].

  1. Reviewer: One suggested line edit in line 240 replace "Stand" with "Standing"

Reply: Ok, we changed as suggested.

Reviewer 2 Report

Comments and Suggestions for Authors

This is a serology paper dealing with a convenience sample of Tuscany adult inhabitants for VZV infections, with their characteristics problems. The manuscript is well written and extensively analyzed, but without information on the vaccination status of younger populations, or herpes zoster vaccination in older populations. This is the main problem, which could be easily corrected by inserting adequate explanations. 

Another problem is the absence of correlation of the reported prevalence of childhood varicella according to age groups, that could explain the higher prevalence in older age groups, if the varicella incidence had declining in the last century. 

Author Response

Response to reviewer 2:

Comments and Suggestions for Authors

  1. Reviewer: This is a serology paper dealing with a convenience sample of Tuscany adult inhabitants for VZV infections, with their characteristic’s problems. The manuscript is well written and extensively analyzed, but without information on the vaccination status of younger populations, or herpes zoster vaccination in older populations. This is the main problem, which could be easily corrected by inserting adequate explanations. 

Reply: Thank you for the comment. We agree with the reviewer that lack of information about immunization status poses a challenge in interpreting our results. This limitation was also discussed in the paper in the following sentence: “As regards limitations, our study did not explore potential factors influencing the observed seroprevalence across different groups (such as assessing vaccination status). As a matter of fact, vaccination determinants have not been investigated, therefore it is not possible to trace the reasons why some subjects are susceptible”.

Indeed, vaccination status for the adult population is not available at either the regional or national level because vaccination data were not electronically collected for adults in the vaccination register in the previous years and even currently there is a lack of this information in Italy. Only self-reported vaccination status against varicella could have been collected at the enrolment with a certain level of uncertainty due to bias recall, especially for older subjects. We think that data collected in such a way would not be reliable, especially for varicella vaccination in adulthood since it was introduced in Tuscany Region in 2008 and at the national level in 2017 and only few cohorts of children have been vaccinated. On the other hand, vaccination against Herpes Zoster has only been recommended at the national level for adults aged 65 years and for certain high-risk groups, but coverage rates were not available for either older people or for high-risk subjects.

According to your suggestion, we added the following sentence in the discussion section:

Line 239-250: For the aim of this investigation, adult sera samples were collected, ten years after the UVV introduction at the regional level, when only few cohorts of children have been vaccinated. The high seroprevalence rates in adulthood observed in our sample can be largely explained by the historically high incidence rates of varicella in Italy. Indeed, adults in Tuscany should have acquired their immunity almost exclusively by natural infection due to the exposure to a massive circulation of varicella zoster virus during the previous decades rather than by vaccination against varicella or Herpes Zoster. On the other hand, vaccination against Herpes Zoster was recommended at the national level for adults aged 65 years and for certain high-risk groups, but coverage rates are not available for either older people or for high-risk subjects.

  1. Reviewer: Another problem is the absence of correlation of the reported prevalence of childhood varicella according to age groups, that could explain the higher prevalence in older age groups, if the varicella incidence had declining in the last century. 

Reply: Thank you for the comment. Our previous reply may partially answer to this comment; however, we acknowledge that while in the last century varicella incidence had declined, underreporting for varicella has been recognized as a persistent problem for the incidence data reliability. For this reason, seroprevalence data are more suitable for a comprehensive understanding of the varicella virus circulation.

According to yours and another reviewer’s comments, we added the following sentence, with reference for varicella underreporting in Italy.

Line 262- 267: Moreover, underreporting is another problem for the reliability of incidence data, as in Italy the level of under notification for varicella, is very high and we can expect the highest underreporting rates in adults, as reported by Ciofi degli Atti et al. [Ciofi degli Atti ML, Rota MC, Mandolini D, Bella A, Gabutti G, Crovari P, Salmaso S. Assessment of varicella underreporting in Italy. Epidemiol Infect. 2002 Jun;128(3):479-84. doi: 10.1017/s0950268802006878. PMID: 12113493; PMCID: PMC2869845]. Therefore, seroprevalence studies are more reliable than incidence data for a comprehensive understanding of the varicella virus circulation and the impact of the vaccination strategies.

Reviewer 3 Report

Comments and Suggestions for Authors

The authors presented an important study on immunity to varicella in Florence pre COVID. In the method section the authors need to explain with a reference why people with an acute infection were excluded as this should not influence already existing antibody levels.

The authors need to present data on documented varicella immunisation and history of clinical varicella in the subjects investigate and put those data in relationship to the antibody test results. 

The conclusion section of the main manuscript needs to be rephrased.

Comments on the Quality of English Language

The authors need to involve a native English speaker in correction of the manuscript.

Author Response

Response to reviewer 3:

Comments and Suggestions for Authors

  1. Reviewer: The authors presented an important study on immunity to varicella in Florence pre COVID. In the method section the authors need to explain with a reference why people with an acute infection were excluded as this should not influence already existing antibody levels.

Reply: We agree with the reviewer that an acute infection should not influence the already existing antibody levels, but we decided not to include subjects who had received a blood transfusion in the previous six months or even subjects who were immunocompromised or taking immunosuppressive treatments.

International literature reported the so-called phenomenon of “immune suppression” occurring for example after measles infection. The severe immune responses induced by measles infection are paradoxically associated with depressed responses to other pathogens, lasting for several weeks to several months or even years beyond resolution of the acute illness [Griffin DE. Measles virus-induced suppression of immune responses. N Engl J Med 2007;357(19):1903-15.]. Therefore, we excluded subjects who have had an acute infectious disease in the previous two weeks as potentially immunocompromised people.

  1. Reviewer: The authors need to present data on documented varicella immunisation and history of clinical varicella in the subjects investigate and put those data in relationship to the antibody test results. 

Reply: Thank you for the comment. We agree with the reviewer that lack of information about immunization status and history of clinical varicella is one problem for the interpretation of our results. On the other hand, the suggestion of the reviewer is not feasible for this study since documented coverage data on varicella vaccination for the adult population are not available in the national and regional registries. Moreover, in Tuscany, vaccination coverage data are available since 2010 (birth cohort 2008) for the paediatric population, but this population is not included in our study.

Data on history of clinical varicella in the enrolled subjects has been collected but they are not showed in the result of the manuscript, as it was not one of the research objectives. In the future maybe we can investigate also those data taking into account that for the adult population reliable notification data are not available and varicella underreporting could be a problem, especially in case of adult population in which the highest underreporting rates are observed in Italy [[Ciofi degli Atti ML, Rota MC, Mandolini D, Bella A, Gabutti G, Crovari P, Salmaso S. Assessment of varicella underreporting in Italy. Epidemiol Infect. 2002 Jun;128(3):479-84. doi: 10.1017/s0950268802006878. PMID: 12113493; PMCID: PMC2869845].]

According to your comment, we have added two new sentences to the discussion:

Line 239-250: For the aim of this investigation, adult sera samples were collected, ten years after the UVV introduction at the regional level, when only few cohorts of children have been vaccinated. The high seroprevalence rates in adulthood observed in our sample can be largely explained by the historically high incidence rates of varicella in Italy. Indeed, adults in Tuscany should have acquired their immunity almost exclusively by natural infection due to the exposure to a massive circulation of varicella zoster virus during the previous decades rather than by vaccination against varicella or Herpes Zoster. On the other hand, vaccination against Herpes Zoster was recommended at the national level for adults aged 65 years and for certain high-risk groups, but coverage rates were not available for either older people or for high-risk subjects.

Line 262- 267: Moreover, underreporting is another problem for the reliability of incidence data, as in Italy the level of under notification for varicella, is very high and we can expect the highest underreporting rates in adults, as reported by Ciofi degli Atti et al [Ciofi degli Atti ML, Rota MC, Mandolini D, Bella A, Gabutti G, Crovari P, Salmaso S. Assessment of varicella underreporting in Italy. Epidemiol Infect. 2002 Jun;128(3):479-84. doi: 10.1017/s0950268802006878. PMID: 12113493; PMCID: PMC2869845]. There-fore, seroprevalence studies are more reliable than incidence data for a comprehensive understanding of the varicella virus circulation and the impact of the vaccination strategies.

  1. Reviewer: The conclusion section of the main manuscript needs to be rephrased.

Reply: We rephrased the conclusion section as follows:

Line 281-294:

Standing on our results, we confirm the importance of continuous varicella monitoring antibody seroprevalence rates and strengthening the current vaccination programs to guarantee adequate levels of immunity in the adult susceptible population, especially in women of childbearing age, to reduce the risk of infection during pregnancy and finally to promote the interruption of varicella transmission. Many efforts should be made in Italy to increase availability of vaccine data for the adult population, since vaccine data were not collected in the vaccination electronic register. Awareness on the need to monitor adult immunization coverage is also being raised at the European level through international initiatives involving experts in vaccine preventable diseases, public health and immunology [Pattyn J, Del Riccio M, Bechini A, Hendrickx G, Boccalini S, Van Damme P, Bonanni P. The Adult Immunization Board (AIB): A new platform to provide multidisciplinary guidelines for the implementation and optimization of adult immunization in Europe. Vaccine. 2024 Jan 1;42(1):1-3. doi: 10.1016/j.vaccine.2023.11.060. Epub 2023 Dec 3. PMID: 38044243.]. Moreover, the ageing population is pushing for easier access to vaccinations to contrast the low adherence to recommended vaccinations for adults and to increase the quality of life, especially for older people [Antonelli-Incalzi R, Blasi F, Conversano M, Gabutti G, Giuffrida S, Maggi S, Marano C, Rossi A, Vicentini M. Manifesto on the Value of Adult Immunization: "We Know, We Intend, We Advocate". Vaccines (Basel). 2021 Oct 22;9(11):1232. doi: 10.3390/vaccines9111232. PMID: 34835163; PMCID: PMC8625332.].

Comments on the Quality of English Language

The authors need to involve a native English speaker in correction of the manuscript.

Reply: We revised the quality of English language.

Round 2

Reviewer 3 Report

Comments and Suggestions for Authors

The authors have made responded to the comments adequately.